# Estrogenic, Antiestrogenic and Antiproliferative Activities of *Euphorbia bicolor* (*Euphorbiaceae*) Latex Extracts and Its Phytochemicals

**DOI:** 10.3390/nu12010059

**Published:** 2019-12-25

**Authors:** Paramita Basu, Elizabeth Meza, Michael Bergel, Camelia Maier

**Affiliations:** Department of Biology, Texas Woman’s University, Denton, TX 76204-5799, USA; pbasu@twu.edu (P.B.); emeza2@twu.edu (E.M.)

**Keywords:** breast cancer, *Euphorbia bicolor*, resiniferatoxin, rutin, latex, MCF-7, T47-D, MDA-MB-231, MDA-MB-469

## Abstract

Estrogen receptor antagonists are effective in breast cancer treatment. However, the side effects of these treatments have led to a rise in searching for alternative therapies. The present study evaluated the estrogenic, antiestrogenic, and antiproliferative activities of *Euphorbia*
*bicolor* (*Euphorbiaceae*), a plant native to south-central USA. Estrogenic and antiestrogenic activities of latex extract and its phytochemicals were evaluated with a steroid-regulated yeast system expressing the human estrogen receptor α and antiproliferative properties were assessed in the ER-positive MCF-7 and T47-D and triple-negative MDA-MB-231 and MDA-MB-469 breast carcinomas. Genistein and coumestrol identified in the latex extract induced higher estrogenic and antiestrogenic activities compared to diterpenes and flavonoids. The latex extract, resiniferatoxin (RTX) and rutin induced antiproliferative activities in all cell lines in a dose-dependent manner, but not in human normal primary dermal fibroblast cultures. A biphasic effect was observed with MDA-MB-468 breast carcinoma in which the latex extract at low concentrations increased and at high concentrations decreased cell proliferation. Treatments with latex extract in combination with RTX or rutin reduced even more the proliferation of MCF-7 breast carcinoma compared to the individual latex, RTX, and rutin treatments. *E. bicolor* latex phytochemicals could contribute to developing commercial therapeutic agents for breast cancer treatment.

## 1. Introduction

Breast cancer is one of the most frequently diagnosed cancers in women. In 2018, 25.4% of the total number of new cases diagnosed were breast cancer [1]. According to the National Cancer Institute, 268,600 (15.2%) new cases of breast cancer have been estimated so far in 2019 [2]. Estrogen receptors, present in approximately 75% of breast cancers, are known to play important roles in the initiation and progression of breast cancer ([3]; and references within). Estrogen receptor alpha especially is an important target for drug development for the prevention and treatment of breast cancer [4] and estrogen receptor positive (ER+) tumors show relatively better prognoses for treating breast cancer as compared to ER-negative breast cancer, especially the triple negative breast cancer, which does not express the hormone epidermal growth factor receptor 2 (HER-2), estrogen receptors (ER), and progesterone receptors (PR) [5].

The biological activities of estrogens are mediated by both genomic and non-genomic pathways [6]. Through the genomic pathway, estrogen binds to the estrogen receptors (ER), which then translocate to the nucleus, where the estrogen-receptor complex interacts with the estrogen response elements (ERE) present in the promoters of the target genes, recruits co-activators, and initiates transcription of those genes. The effects of estrogen are also mediated via DNA-binding transcription factors (activator protein-1, nuclear factor-κB, and stimulating protein-1), which mediate ER association with the DNA. Estrogen also elicits its activity via different non-genomic signaling pathways, such as NF-κB, PI3K/Akt/mTOR, and MAPK/ERK associated with growth factor activation [7]. ERα, present in approximately 75% of breast cancers [8], is known to play important roles in the initiation and progression of breast cancer [9]. Therefore, ERα is an important target for drug development for the prevention and treatment of breast cancer [10]. 

Breast cancers are mainly treated with surgery, radiotherapy, chemotherapy, SERMs (tamoxifen), estrogen receptor down-regulators (fulvestrant), aromatase inhibitors (anastrozole, exemestane, letrozole), several drugs (ado-trastuzumab emtansine, lapatinib, palbociclib, pertuzumab, and trastuzumab), antimetabolites (methotrexate), DNA-interactive agents (cisplatin, doxorubicin), antitubulin agents (taxanes), etc. [11,12]. However, the clinical uses of the drugs are accompanied by several side effects, such as neurological dysfunction, hair loss, drug resistance, bone marrow suppression, gastrointestinal lesions, etc. [11,13,14]. Therefore, researchers have focused on the search for new anticancer agents with better efficacy and less side effects. Several studies have reported plant-derived bioactive compounds with anticancer properties that inhibit cell cycle regulators, cell proliferation, angiogenesis, metastasis and different signaling pathways (NF-κB, PI3K/Akt/mTOR, and MAPK/ERK), induce apoptosis, modulate estrogen biosynthesis and metabolism, and reverse multidrug resistance ([15]; and references within). Natural products have long been a prominent source of anti-cancer drugs, representing 48.6% of those FDA-approved drugs since 1940 [16].

*Euphorbiaceae* species are known to possess anticancer properties by inducing apoptosis and cell cycle arrest [17,18], inhibiting cell proliferation [19], and also reversing the multidrug resistance [20] of different breast cancer cell lines. Snow-on-the-prairie, *Euphorbia bicolor* (Spurge family, *Euphorbiaceae*), a species native to south-central USA have not been studied before. The present study evaluated the estrogenic, antiestrogenic and antiproliferative properties of *E*. *bicolor* (*Euphorbiaceae*) latex extract and its phytochemicals. We report that the *E*. *bicolor* latex extract and its phytochemicals induced estrogenic and antiestrogenic properties in a steroid-regulated yeast system and also possessed antiproliferative properties in ER-positive MCF-7 and T47-D and triple negative MDA-MB-231 and MDA-MB-469 breast carcinomas without affecting the growth of the human normal primary dermal fibroblast cell line. To our knowledge, this is the first study reporting the estrogenic, antiestrogenic, and antiproliferative properties of *E*. *bicolor* latex extract and resiniferatoxin (TRX) on ER-positive T47-D and triple-negative MDA-MB-469 breast carcinomas.

## 2. Materials and Methods

### 2.1. Plant Collection and Preparation of Latex Extract

*E. bicolor* plants were collected from prairies in Denton County, Texas, USA. The extracts were prepared by the same method as Basu et al. [21]. Fresh latex was collected from different plant organs in pre-weighed vials and extracted in 80% methanol (1:40 *w*/*v*) at room temperature for two days. The extract was centrifuged at 3500 rpm for 20 minutes and the supernatant was filtered through Whatman #54 filter paper (Thomas Scientific, Swedesboro, NJ, USA) and stored at −20 °C for future use.

### 2.2. Identification of Latex Phytochemicals by UPLC-ESI-MS/MS

Latex phytochemicals were identified by employing ultra-performance liquid chromatography electrospray ionization tandem mass spectrometry (UPLC-ESIMS/MS) [21]. A Waters Acquity UPLC (Waters Corporation, Milford, MA, USA) chromatography system, coupled with ESI Xevo TQD triple quadrupole mass spectrometer, was used for phytochemicals identification. Latex phytochemicals were separated in a positive ionization mode in a Restek Raptor biphenyl column (100 mm length × 2.1 mm diameter × 1.8 micrometer particle size). Chromatographic analyte separations were carried out using a gradient mobile phase consisting of 0.1% formic acid in 10 mM ammonium formate and 0.1% formic acid in acetonitrile under linear gradient conditions (A:B % *v*/*v*, 0–0.5 min: 80:20; 0.5–14 min: 30:70; 14–15 min: 80:20) at 0.6 mL/min flow rate. The column temperature was maintained at 50 °C. Cone gas flow of 10 L/h, desolvation gas flow of 1000 L/h, capillary voltage of 0.70 kV, source temperature of 150 °C, and desolvation temperature of 450 °C were maintained for the source-dependent parameters. Ion detection was performed in the multiple-reaction monitoring (MRM) mode by monitoring the transition pairs. The compounds were identified based on standards prepared in methanol.

### 2.3. Estrogenic and Antiestrogenic Assays

A steroid-regulated *Saccharomyces cerevisiae* system, BJ3505 [MAT a, pep 4: His 3, prb l–Δ 1.6 R, his 3–Δ 200, lys 2-801, trp l–Δ 101(gal3), ura 3-52(gal 2), can 1], containing a human ERα expression plasmid and a β-galactosidase gene reporter plasmid was used to determine the estrogenic and antiestrogenic activities of latex extract and its phytochemicals [22,23]. The estrogenic and antiestrogenic assays were performed by the modified method of Maier et al. [24]. In brief, stock yeast cultures were grown in a casamino acid–glucose medium (20% glucose, 10% YNB, and 5% adenine sulfate) at 230 rpm, 30 °C, overnight in an incubator-shaker. An estradiol (E) standard curve was used to estimate the estrogen equivalents (E equiv) in the latex extract. Ten milliliter of yeast subcultures were inoculated with 100 µg, 200 µg, 300 µg, or 400 µg E equivalents of *E*. *bicolor* latex extract or its identified phytochemicals at concentrations of 0.5, 1, and 5 µM. Cultures were grown for 6 hours after inoculation with latex extract or phytochemicals. Estradiol and genistein were used as positive controls. Yeast cultures without any treatments were used as negative controls. Estrogenic activity experiments were repeated three times and each experiment had two replicates.

For antiestrogenic assays, the yeast cultures were inoculated with estradiol and 100 µg, 200 µg, 300 µg, or 400 µg E equivalents of latex extract or phytochemicals resiniferatoxin (RTX) and rutin at concentrations of 0.5, 1, and 5 µM. The antiestrogenic activity translates in a reduction of the transcriptional activity induced by estradiol in transgenic yeast due to the interference or competition of latex extract and phytochemicals with estradiol for the ligand-binding site of the ER. Antiestrogenic activity experiments were performed with two replicates each repeated three times.

Yeast cells were disrupted with glass beads (0.5 mm diameter; BioSpec Products, Inc., Bartlesville, OK, USA) and the protein concentrations were estimated spectrophotometrically using the Bio-Rad protein assay dye reagent (BioRad, Berkeley, CA, USA). β-galactosidase assays were performed by incubating estimated amount of protein in Z-buffer (0.006 M·Na_2_HPO_4_·7H_2_O, 0.04 M NaH_2_PO_4_·H_2_O, 0.01 M KCl, 0.001 M·MgSO_4_, and 0.05 M β-mercaptoethanol), and 4 mg/mL *ortho*-Nitrophenyl-β-galactoside (ONPG). The reaction mixtures were incubated at room temperature for 30 minutes and terminated by adding 500 µL of 1M Na_2_CO_3_. The β-galactosidase activity of each reaction was measured spectrophotometrically at 420 nm. Estrogenic activities of the latex extract and phytochemicals were expressed in Miller Units (MU) (Equation (1)).
(1)$$MU=1000\;\times \frac{\left(Abs\;at\;420\;nm\right)}{Protein\;Concentration\;\left[g\right]\;\times \;Reaction\;Time\;\left[min\right]}\;$$

Antiestrogenic activities of the latex extract and phytochemicals were calculated according to Equation (2) and expressed as % inhibition of estradiol activity.
(2)$$MU=100-1000\;\times \frac{\left(Abs\;at\;420\;nm\right)\;}{Protein\;Concentration\;\left[g\right]\;\times \;Reaction\;Time\;\left[min\right]}$$

Abs = Absorbance

### 2.4. Cell Lines and Cell Culture Conditions

Estrogen receptor positive MCF-7 and T-47D, triple negative MDA-MB-231 and MDA-MB-468 breast cancer cell lines, and adult human normal primary dermal fibroblast were obtained from American Type Culture Collection (ATCC, Manassas, VA, USA). MCF-7 and MDA-MB-231 breast carcinomas were maintained in Dulbecco’s Modified Eagle’s Medium (DMEM; ThermoFisher Scientific, Township, NJ, USA), supplemented with 10% heat-inactivated fetal bovine serum (Gemini), 1% penicillin and streptomycin (Gibco/BRL). Cells were grown in a humidified atmosphere of 5% CO_2_ at 37 °C. T-47D cells were maintained in RPMI-1640 medium, (ATCC), containing 0.2 Units/ml bovine insulin; 10% fetal bovine serum and human insulin (Life Technologies, Carlsbad, CA, USA). MDA-MB-468 cells were maintained in Leibovitz’s L-15 medium (ATCC), containing 10% fetal bovine serum and in absence of CO_2._ All cell lines were maintained in the logarithmic growth phase by routine passage every 2–3 days using 0.025% trypsin-EDTA treatment. After 75%–80% confluency, the cells were trypsinized and transferred to a DMEM, high glucose, no glutamine, no phenol red free medium (Gibco, Life Technologies), supplemented with sodium pyruvate (Gibco, Life Technologies) and GlutaMAX™ supplement (Gibco, Life Technologies). Cells were maintained in phenol-red free medium until they reached 75%–80% confluency and then seeded in 96-well plates. Human normal primary dermal fibroblast cells were grown in fibroblast basal medium (ATCC) supplemented with fibroblast growth kit-low serum (ATCC). Cells were grown in 5% CO_2_ at 37 °C.

### 2.5. Cell Culture Treatments

All cells were seeded into 96-well cell culture plates at 9 × 10^3^ cells/well and incubated for 24 h at 37 °C. After 24 h, the cells were treated with different concentrations of *E*. *bicolor* latex extract (1.96 µg/mL, 3.91 µg/mL, 7.81 µg/mL, 15.63 µg/mL, 62.5 µg/mL, 125 µg/mL, 250 µg/mL, 500 µg/mL) and its identified phytochemicals RTX and rutin at 0.5 µM, 1 µM, 5 µM, 10 µM, 50 µM, 100 µM, 250 µM, and 500 µM concentrations. The extract and its phytochemicals were re-suspended in less than 0.1% DMSO. Human normal primary dermal fibroblast cells were treated with the abovementioned concentrations of latex extract or phytochemicals RTX or rutin.

In another series of experiments, latex extract and RTX or rutin were added individually or in combination to MCF-7 cells. The final concentration of DMSO was <0.5%. After treatment with the combination of latex extract and its phytochemicals, the MCF-7 cells were grown at 37 °C for 72 h. Antiproliferative activity was evaluated by preforming MTS [3-(4,5-dimethylthiazol-2-yl)-5-(3-carboxymethoxyphenyl)-2-(4-sulfophenyl)-2H-tetrazolium] assay (Promega, Madison, WI, USA). The medium was removed and fresh medium with MTT was added to each well for 3 h at 37 °C. Plate readings were taken at 490 nm using a Biotek’s Synergy HT plate reader. Antiproliferative assays were performed as three independent experiments each with three replicates.

### 2.6. Statistical Analyses

Means and standard errors of the mean were calculated. One-way ANOVA followed by Tukey’s posthoc test were performed to determine the significant differences (*p* ≤ 0.05) among the means for the antiproliferative assays using GraphPad Prism 7. Estrogenic and antiestrogenic activities were analyzed by two-way ANOVA followed by Bonferroni’s posthoc at *p* ≤ 0.05. Growth inhibition (GI_50_) values were calculated by linear regression analysis.

## 3. Results

### 3.1. Estrogenic Activities of Latex Extract and Its Phytochemicals in a Steroid-Regulated Yeast System

To test the estrogenic activities of *E*. *bicolor* latex extract and its identified phytochemicals, a steroid-regulated yeast system expressing ERα was employed. Latex extract in the range of 100 E–400 E equivalents induced low, not dose-dependent estrogenic activities ranging from 115.3 ± 6.3 to 121.7 ± 7.8 MU (Figure 1a). Coumestrol (Figure 1b) and genistein (Figure 1d) significantly induced estrogenic activities in a dose-dependent manner at 0.5, 1, and 5 µM. Abietic acid (84.4 ± 13.1 MU) (Figure 1b), RTX (88.3 ± 11.9 MU) (Figure 1b), and daidzein (87.4 ± 10.9 MU) (Figure 1d) showed low estrogenic activities at all concentrations tested. Estrogenic activities of all flavonoids identified in latex extract (chalcone, kaempferol, naringenin, quercetin, and rutin) were 90–100 MU at all concentrations (Figure 1c). No significant differences were observed among the estrogenic activities induced by all concentrations (0.5, 1, 5 µM) of flavonoids chalcone, kaempferol, naringenin, quercetin, and rutin tested in the present study (Figure 1c).

### 3.2. Antiestrogenic Activities of Latex Extract and Its Phytochemicals in a Steroid-Regulated Yeast System

To test for antiestrogenic activity, yeast cells were treated with estradiol and *E*. *bicolor* latex extract or its identified phytochemicals. The latex extract induced antiestrogenic activity by dose-dependently inhibiting the estradiol activity in the steroid-regulated yeast system. At 400 E equivalents, the latex extract inhibited estradiol activity by 50% (Figure 2a). Coumestrol (Figure 2b) and the isoflavones daidzein and genistein (Figure 2d) showed significantly higher antiestrogenic activities compared to other latex phytochemicals by inhibiting the estradiol activity by 70.7%, 70%, and 90.2%, respectively. Antiestrogenic activities of all flavonoids identified in latex extract, chalcone, kaempferol, naringenin, quercetin, and rutin were 10.3%–44% at all concentrations (0.5, 1, and 5 µM) (Figure 2c). No significant differences were observed among the antiestrogenic activities induced by all concentrations of the flavonoids chalcone, quercetin, kaempferol, rutin and naringenin tested in the present study (Figure 2c).

### 3.3. Antiproliferative Activities of Latex Extract and Its Phytochemicals on ER-Positive and Triple Negative Breast Cancer Cell Lines

To test the antiproliferative activities of *E*. *bicolor* latex extract and its identified phytochemicals RTX and rutin, ER-positive MCF-7 and T47-D and triple negative MDA-MB-231 and MDA-MB-468 breast carcinomas were employed. The *E*. *bicolor* latex extract dose-dependently inhibited the proliferation of both ER-positive cell lines, MCF-7 and T47-D (Figure 3a,b). At 500 µg/mL, the proliferation of MCF-7 and T47-D cells were significantly reduced by 61.1% and 65%, respectively (Figure 3a,b). The GI_50_ of the latex extract for MCF-7 and T47D cell lines were 498.7 ± 1.3 µg/mL and 315.7 ± 36.6 µg/mL, respectively (Table 1). Latex extract dose-dependently inhibited the proliferation of both triple negative cell lines, MDA-MB-231 and MDA-MB-468. At 500 µg/mL, the proliferation of MDA-MB-231 and MDA-MB-468 were significantly reduced by 72.8% and 54.4%, respectively (Figure 3c,d). Latex extract showed a significant biphasic effect in MDA-MB-468 cells. The proliferation of MDA-MB-468 cells was significantly increased at 1.96 µg/mL–15.63 µg/mL of latex extract treatment, whereas the increasing extract concentrations (62.5 µg/mL–500 µg/mL) inhibited its proliferation (Figure 3d). The GI_50_ of the latex extract for MDA-MB-231 and MDA-MB-468 cell lines were 258.3 ± 18 µg/mL and 499 ± 0.8 µg/mL, respectively (Table 1). *E*. *bicolor* latex extract at concentrations up to 62.5 µg/mL did not inhibit the growth of human normal primary dermal fibroblast. A decrease of up to 83% in cell viability was observed with higher concentrations of latex extract treatment (Figure 3e).

Phytochemicals RTX and rutin identified in the latex extract also dose-dependently inhibited the proliferation of the ER-positive cell lines. RTX significantly reduced the proliferation of MCF-7 cells by 91% both at 250 µM and 500 µM and reduced the proliferation of T47-D cells by 85.1% and 95.7% at 250 µM and 500 µM, respectively (Figure 4a,b). At 500 µM, rutin significantly reduced the proliferation of MCF-7 and T47-D cells by 83.3% and 94%, respectively (Figure 5a,b). The GI_50_ of RTX for MCF-7 and T47D cell lines were 139 ± 7.8 µM and 100 ± 23.6 µM, respectively (Table 1). The GI_50_ of rutin for MCF-7 and T47D cell lines were 77.5 ± 18.8 µM and 65.7 ± 14 µM, respectively (Table 1). At 500 µM, RTX significantly reduced the proliferation of triple negative cell lines MDA-MB-231 and MDA-MB-468 cells by 97.7% and 84%, respectively (Figure 4c,d). Rutin also dose-dependently inhibited the proliferation of triple negative cell lines. At 500 µM, rutin significantly reduced the proliferation of MDA-MB-231 and MDA-MB-468 cells by 66% and 70%, respectively (Figure 5c,d). The GI_50_ of RTX for MDA-MB-231 and MDA-MB-468 cell lines were 246.7 ± 3.4 µM and 248.5 ± 1.5 µM, respectively (Table 1). The GI_50_ of rutin for MDA-MB-231 and MDA-MB-468 cell lines were 160 ± 8.2 µM and 383.3 ± 54.4 µM, respectively (Table 1). The same significant biphasic effect of RTX and rutin in MDA-MB-468 cells as for the latex extract treatment was observed. The proliferation of MDA-MB-468 cells was significantly increased by lower concentrations, and decreased by higher concentrations (250 µg/mL and 500 µg/mL) of RTX and rutin (Figure 4d and Figure 5d). RTX inhibited the growth of human normal primary dermal fibroblast cells up to 82% at higher concentrations (Figure 4e). Rutin did not significantly inhibit human normal primary dermal fibroblast cell viability (Figure 5e).

### 3.4. Antiproliferative Activity of Latex Extract in Combination with RTX or Rutin in MCF-7 Cell Line

To test for synergistic and additive effects, latex extract-RTX and latex extract-rutin, combination treatments of MCF-7 cells were assayed. The latex extract-RTX combination significantly reduced the proliferation of MCF-7 cells at all concentrations by 51%–80% (Figure 6a) as compared to the individual treatment of only latex extract (Figure 6e) or RTX (Figure 6c). Similarly, the treatment of latex extract-rutin combination significantly reduced the proliferation of MCF-7 cells at all concentrations by 31.7%–78.2% (Figure 6b) as compared to the individual treatment of only latex extract (Figure 6e) or rutin (Figure 6d).

## 4. Discussion

*E*. *bicolor* latex extract and its phytochemicals showed dose-dependently antiestrogenic activities in a steroid-regulated yeast system, and induced significant antiproliferative activities in ER-positive and triple negative breast cancer cells-but not in human normal primary dermal fibroblast. In vitro yeast two hybrid system containing ERα or ERβ has been used previously as a potential tool for screening for estrogenic and antiestrogenic activities of several plant species [23,24,25,26]. The present study reports low estrogenic activity of the isoflavone daidzein and flavonoids chalcone, kaempferol, naringenin, quercetin, and rutin identified in *E*. *bicolor* latex extract. The low estrogenic activity of daidzein could be explained by the 1000 times lower affinity of daidzein for estrogen receptor compared to that of estradiol [27]. Resende et al.’s study showed that among a series of flavonoids tested, only kaempferol induced significant estrogenic activity in a recombinant yeast assay. Quercetin, fisetin, chrysin, luteolin, galangin, flavone, 3-hydroxyflavone, 5-hydroxyflavone and 7-hydroxyflavone showed no detectable estrogenic activity [28]. In our study, coumestrol (1–5 µM) and genistein (1–5 µM) induced high estrogenic activities. A previous study showed that coumestrol at 1 µM and genistein at 10 µM induced estrogenic activities comparable to the estrogenic activity of estradiol at 0.01 µM in a yeast estrogen screen assay [29].

In the current study, 5 µM of coumestrol, daidzein, and genistein, known phytoestrogens, induced antiestrogenic activities by inhibiting 70.7%, 69.2%, 90.3% of estradiol activity, respectively. Chalcone, kaempferol, naringenin, quercetin, and, abietic acid, RTX, and rutin, identified in the *E*. *bicolor* latex extract, also induced much lower antiestrogenic activities. It has been reported that coumestrol and genistein did not exhibit antiestrogenic activities, whereas biochanin A and naringenin induced antiestrogenic activities at a nanomolar range in yeast estrogen screen assay [29]. The differences between the antiestrogenic activities we observed and those obtained by Collins et al. could be attributed to the employment of different yeast systems and cultural conditions [29]. The results of strong antiestrogenic activities of phytoestrogens, mainly of genistein, corroborate with their antiproliferative activity. Genistein has been reported to enhance hormonal therapy sensitivity in MDA-MB-231 [30] and induce apoptosis and autophagy in MCF-7 breast carcinomas [31].

In the present study, *E*. *bicolor* latex extract reduced the proliferation of both ER-positive (MCF-7 and T47-D) and triple negative (MDA-MB-231 and MDA-MB-468) breast cancer cell lines. The *Euphorbiaceae* family species are well-known for their antiproliferative properties. It was previously reported that *E. hirta* methanolic extract significantly inhibited the proliferation of MCF-7 cells at 24 h at the GI_50_ value of 25.26 µg/mL [32], much lower than that of *E. bicolor* latex extract in our study. Acetone extracts of *E. macroclada* leaves and flowers induced significant cytotoxicity in MCF-7 breast carcinoma [33]. Butanol, hexane or methanol extracts of *E. tirucalli* stem inhibited the proliferation of MCF-7 and MDA-MB-231 in a concentration dependent manner [19]. In another study, dichloromethane and ethyl acetate extracts of *E. macroclada* exhibited cytotoxic effects on MDA-MB-468 cancer cell line, while the methanol extract and latex extract in DMSO were not cytotoxic at the tested concentrations [34]. Sadeghi-Aliabadi et al. reported that the nonpolar extracts of *E. macroclada* possess higher cytotoxic activity than the polar extracts [34]. In our study, the polar methanolic extract of *E*. *bicolor* latex possessed antiproliferative activities on both ER-positive and triple negative breast carcinomas. Other species of *Euphorbia* were reported to possess antiproliferative activities against different breast carcinomas, such as *E*. *szovitsii* in MDA-MB-231 [18], *E*. *humifusa* in MCF-7, T47-D, and MDA-MB-231 [35], and *E*. *fischeriana* in MCF-7 [36]. The results of above-mentioned studies show that different solvent extracts of *Euphorbia* species and different plant tissues possess antiproliferative activities against different breast cancer cell lines. Various solvents extract different classes of *Euphorbia* phytochemicals and that explains differences in the antiproliferative activities induced in breast cancer cell lines. Thus, differences in the degree of antiproliferative activity results between our and other studies could be attributed to the presence of different groups of phytochemicals in extracts and solvent extraction procedures of different plant tissues. Among the above cited studies, only few report identified chemicals in the *Euphorbia* species and most of those chemicals are flavonoids and terpenoids [17,19,36], as we identified in *E. bicolor* latex extract. Obtusifoliol-related steroids also were identified in *E. sogdiana* [17].

The present study showed that RTX significantly inhibited the proliferation of all breast carcinomas employed in the study. RTX is a potent agonist of the transient receptor potential vanilloid 1 (TRPV1) [37], a non-selective cation channel with high calcium permeability, that is mostly studied in pain research involving the central and peripheral terminals of small diameter sensory neurons [38]. The presence of TRPV1 in MCF-7 breast carcinoma also has been reported [39]. The growth inhibitory activity of RTX in MCF-7 cell cultures was reported previously but no mechanism of action was described [39,40]. RTX also induced apoptosis in the human bladder cancer cell lines T24 and 5637, being associated with mitochondrial dysfunction [41]. As reported by another study [42], in our study we also observed that low micromolar concentrations of RTX treatment showed a trend in increasing the proliferation of all cell lines, including the human normal primary dermal fibroblast cell line, but the increase was significant only for MDA-MB-468. However, at high concentrations, RTX decreased proliferation of all cancer cell lines under study. Pecze et al. reported that at low micromolar concentrations RTX was unable to reduce the cell viability of MCF-7 MDA-MB-231 and BT-474 cells, which was decreased at high RTX doses. The cytotoxic mechanisms of RTX in the above cancer cell lines are based on strong sodium and calcium influx via TRPV1 channels, which lead to mitochondrial calcium accumulation and necrotic cell swelling [42]. The results of RTX treatment of T47-D and MDA-MB-468 breast carcinomas in our study are novel, since no other studies have reported the antiproliferative activity of RTX in these cell lines.

Rutin, a flavonoid glycoside (α-l-rhamnopyranosyl-(1→6)-β-d-glucopyranose), identified in the *E*. *bicolor* latex extract, also showed antiproliferative activities on all four breast cancer cell lines in the present study. Another study has shown that grape stem extract rich in flavonols, particularly rutin, inhibited the growth of MCF-7 cancer cell line [43]. Elsayed et al. reported that rutin inhibited c-Met kinase activity in the triple negative breast cancer cell line, significantly reducing the growth of MDA-MB-231/GFP orthotopic xenograft in nude mouse model [44]. Rutin is a common dietary flavonoid found in fruits, vegetables, and plant-derived beverages [45]. In vivo, gut microflora metabolizes rutin to a variety of compounds, such as quercetin and phenol derivatives [46]. It has been shown that flavonoid metabolites of parent compounds are responsible for antiproliferative activity in colorectal cancer [47]. Moreover, extraction procedures can contribute to rutin transformation. Dawidowicz et al. obtained at least 23 compounds formed from rutin during simulated and real extraction of elderberry flowers with methanol and methanol-water solvents [48]. It is possible that the antiproliferative activities observed in our study are due to such rutin derivatives obtained during *E. bicolor* latex extraction. Since rutin demonstrates beneficial pharmacological activities, such as antioxidant, vasoprotective, anticarcinogen, and neuroprotective properties to mention just a few [45]-more stable, and more absorbent non-toxic analogs and derivatives have been synthesized [49], making them promising potential nutraceuticals.

Many phytoestrogens (flavonoids, stilbenes and cumestans) appear to have a biphasic effect on cell proliferation, stimulating growth at low concentrations and suppressing growth at high concentrations. Genistein was shown to increase growth in estrogen-sensitive cells at low concentrations, but decreased cell growth, suppressed DNA synthesis, and induced cell death at high concentrations [50,51]. A study revealed that the proliferation observed in daidzein-treated MCF7 cells was blocked by the antiestrogen ICI 182,780, indicating that the stimulatory effect exerted by daidzein was ER-mediated [52]. Some of these effects are explained by the interactions of phytoestrogens with ER subtypes, which recruit cofactors, regulate gene expression and stimulate or inhibit the growth of breast carcinomas in different ways [53]. Therefore, in the present study, the biphasic effects of *E. bicolor* latex extract or its phytochemicals RTX or rutin towards different breast carcinomas under study could be explained by the agonistic/antagonistic effects of low vs. high concentrations of the latex extract or its phytochemicals for different ER subtypes. It is also of interest to note the possible role of G protein-coupled estrogen receptors (GPERs) in proliferation, since many phytoestrogens activate GPERs. In particular, genistein has been reported to stimulate growth of MCF-7 cells through a GPER-dependent mechanism [54]. The intake of soy products, rich in genistein, has been attributed to lower incidence of breast cancer in Asian vs. Western populations and epidemiologic studies showed an inverse relation between soy-rich diet and cancer risk [55,56]. Low dose (<10 μmol/L) of genistein has agonistic effects as estrogen activator raising breast cancer risk and high dose (>10 μmol/L) possesses antiproliferative effects ([56] and references within). A clinical study showed that a daily dose of 54 mg of purified genistein did not induce breast or uterine carcinogenicity [57]. However, most clinical studies failed to report a protective effect of high genistein intake against breast cancer risk (reviewed in 56). Therefore, genistein bioactivity is not solely dependent on dose, other factors being involved.

The latex extract and phytochemicals RTX or rutin used in our study did not practically reduce the proliferation of human normal primary dermal fibroblast. Farfariello et al. showed that RTX did not induce cell death in normal human urothelial cells, which supports the findings of the present study [41]. A previous study assessing the effects of rutin on human normal primary dermal fibroblast [58], also supports the findings of our study, thus suggesting that the latex phytochemical rutin targets cancer cells and spares normal cells.

Results from the combination treatments of latex extract with RTX or rutin showed synergistic antiproliferative effect at lower concentrations and additive effects at higher concentrations in MCF-7 cells as compared to individual latex extract or phytochemical treatments. It seems that the latex extract-RTX and latex extract-rutin combinations worked via different pathways to inhibit cell proliferation. Deveci et al. reported that TRPV1-mediated Ca^2+^ entry stimulated chemotherapeutic agent 5-flurouracil induced apoptotic cell death in MCF-7 cells via up-regulation of mitochondrial oxidative stress, caspase, and PARP by a Ca^2+^ signaling molecular pathway [59]. Another study showed that TRPV1 agonist capsaicin enhanced intracellular calcium concentration and thus increased intracellular reactive oxygen species, mitochondrial membrane depolarization, apoptosis, PARP, and caspase activities in MCF-7 carcinomas in response to chemotherapeutic agent doxorubicin and melatonin [60]. Thus, it can be speculated that, in the present study, latex extract-RTX treatment enhanced the antiproliferative activity in MCF-7 cells through the TRPV1-induced Ca^2+^ signaling pathway. Another study showed that rutin acted as a chemosensitizer by increasing the cytotoxicity of two chemotherapeutic drugs, cyclophosphamide and methotrexate, in MDA-MB-231 cancer cells [58]. The role of calcium in the proliferation and invasion of cancer cells, as well as the increased expression of calcium channels in some cancers, led to the assessment of calcium channel inhibitors as potential therapy for cancer [61,62]. It is important to find new chemotherapeutics with unique effective mechanisms of action for treating cancers. Thus, transient receptor potential cation channels (TRP) become cancer-relevant targets for cancer therapies, modulated by natural products as potential anti-cancer agents.

## 5. Conclusions

To the best of our knowledge, this is the first study reporting the estrogenic and antiestrogenic activities of *E*. *bicolor* latex extract and its identified phytochemicals in a steroid-regulated yeast system, and the antiproliferative activities of the latex extract and two of its phytochemicals, RTX and rutin in both ER-positive and triple negative breast carcinomas. The latex extract and its phytochemicals are effective in inhibiting both ER-positive MCF-7 and T47-D and triple negative MDA-MB-231 and MDA-MB-468 human breast cancer cell lines, but not the human normal primary dermal fibroblast cells. Latex extract-RTX and latex-rutin combination treatments showed synergistic effects at low concentrations and additive effects at high concentrations in reducing MCF-7 cell proliferation, indicative of more than one potent phytochemical in the latex and mechanistic pathway. With future studies elucidating the molecular mechanisms of action of the latex phytochemicals in both ER-positive and triple negative breast cancer cell lines, new potential breast cancer therapeutics could be developed. A deeper understanding of TRPV1 channel biology in cancer cell-lines will open opportunities for finding natural products and developing new anti-cancer agents.

## Figures and Tables

**Figure 1 nutrients-12-00059-f001:**
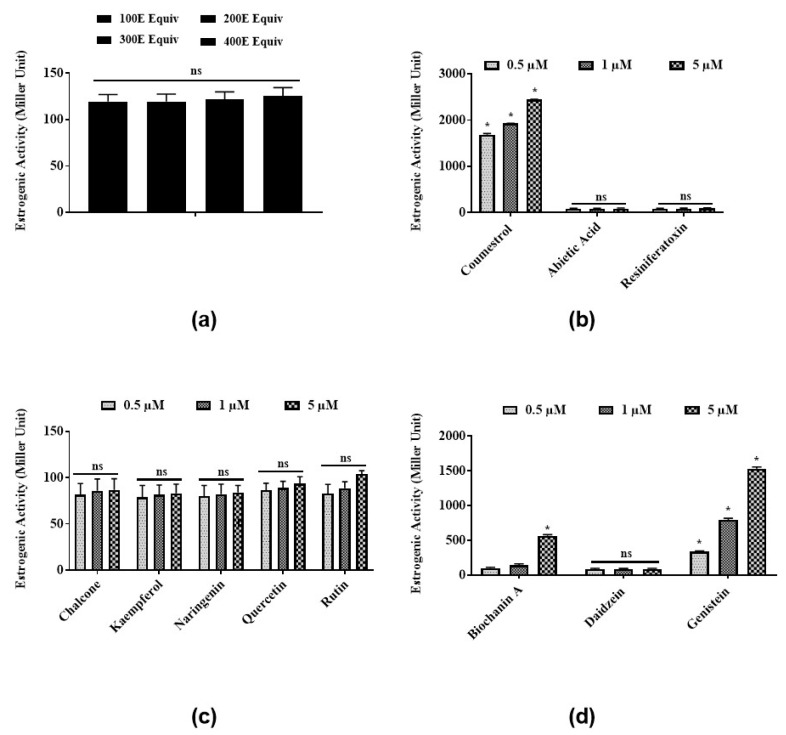
Estrogenic activities of *E*. *bicolor* latex extract and its phytochemicals in a steroid-regulated yeast system (BJ3505): (**a**) *E*. *bicolor* latex extract; (**b**) and its phytochemicals, coumestans (coumestrol) and diterpenes (abietic acid, resiniferatoxin (RTX)); (**c**) flavonoids (chalcone, kaempferol, quercetin, naringenin, quercetin, and rutin); (**d**) and isoflavones (biochanin A, daidzein, genistein). The average activities for positive controls estradiol and genistein were 2353.33 ± 166.6 and 998.9 ± 46.5 MU, respectively. The activity of yeast cultures without any treatments (negative control) ranged from 12.2 ± 0.8 to 24.8 ± 9.3 MU. Estrogenic activity of latex extract was analyzed by one-way ANOVA followed by Tukey’s posthoc test at *p* ≤ 0.05. Estrogenic activity of latex phytochemicals was analyzed by two-way ANOVA followed by Bonferroni posthoc test at *p* ≤ 0.05. Bars with asterisks for each group of chemicals tested are significantly different from each other. ns = non-significant.

**Figure 2 nutrients-12-00059-f002:**
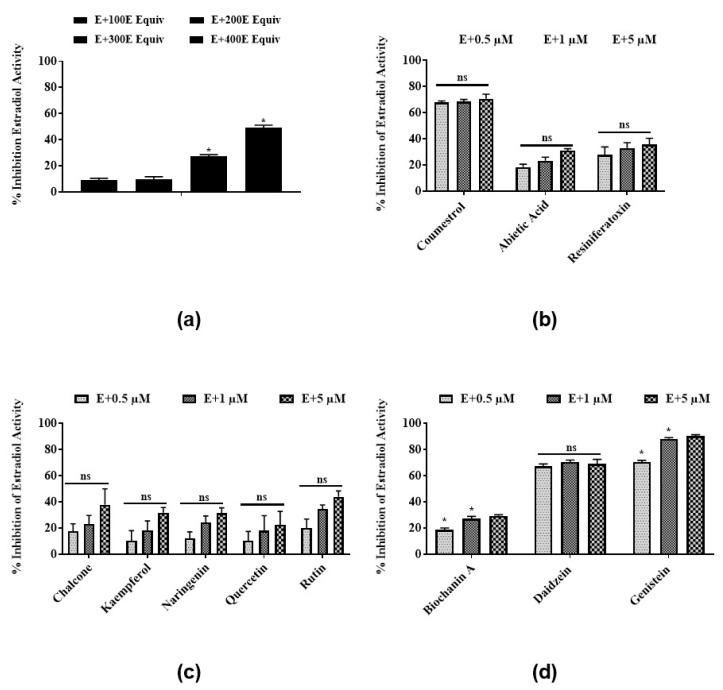
Antiestrogenic activities of *E*. *bicolor* latex extract and its phytochemicals in a steroid-regulated yeast system (BJ3505): (**a**) *E*. *bicolor* latex extract; (**b**) its phytochemicals coumestans (coumestrol) and diterpenes (abietic acid, RTX); (**c**) flavonoids (chalcone, kaempferol, quercetin, naringenin, quercetin, and rutin); (**d**) and isoflavones (biochanin A, daidzein, genistein). The average activity of estradiol (positive control) was 2353.33 ± 166.6 MU. The estrogenic activity of yeast cultures without any treatments (negative control) ranged from 12.2 ± 0.8 to 24.8 ± 9.3 MU. Antiestrogenic activity of latex extract was analyzed by one-way ANOVA followed by Tukey’s posthoc test at *p* ≤ 0.05. Antiestrogenic activity of latex phytochemicals were analyzed by two-way ANOVA followed by Bonferroni posthoc test at *p* ≤ 0.05. Bars with asterisks are significantly different from each other. ns = non-significant.

**Figure 3 nutrients-12-00059-f003:**
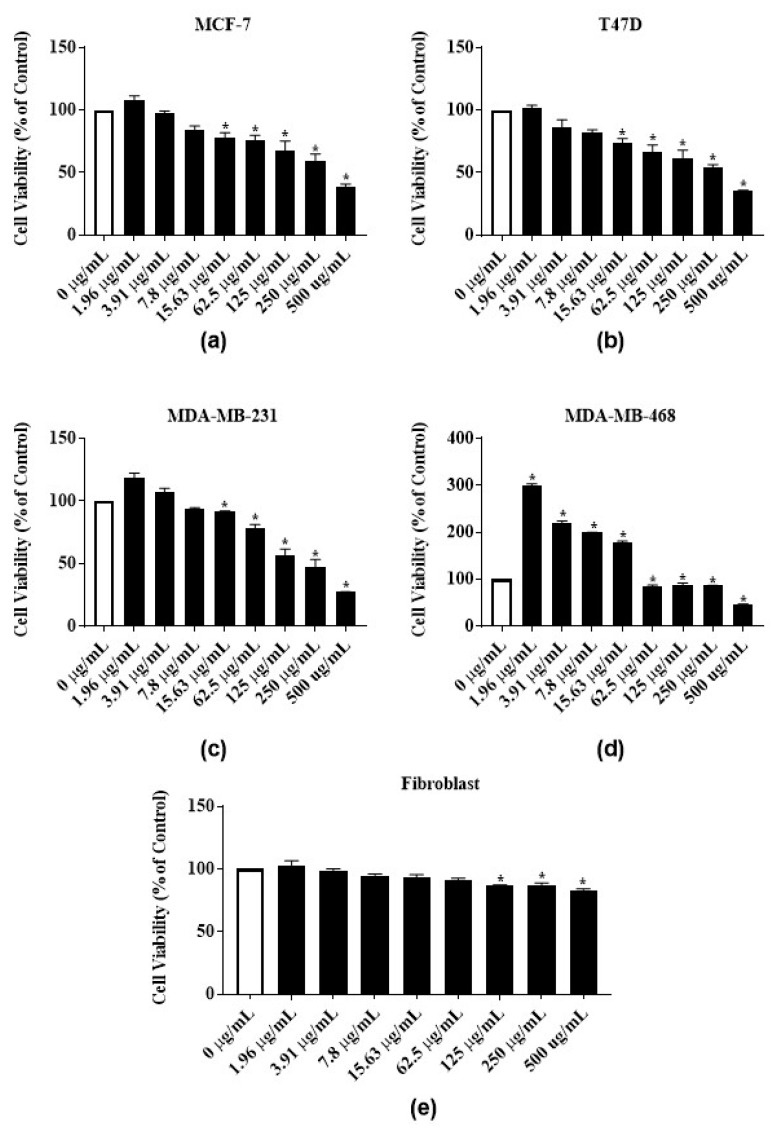
Antiproliferative activities of *E*. *bicolor* latex extract on ER-positive and triple negative breast cancer cell lines: (**a**) MCF-7 (ER+); (**b**) T47-D (ER+); (**c**) MDA-MB-231 (triple negative); (**d**) MDA-MB-468 (triple negative) breast cancer cell lines; and (**e**) effects of *E*. *bicolor* latex on the growth of human normal primary dermal fibroblast. One-way ANOVA followed by Tukey’s posthoc test was performed. Bars with asterisks are significantly different from untreated control at *p* ≤ 0.05.

**Figure 4 nutrients-12-00059-f004:**
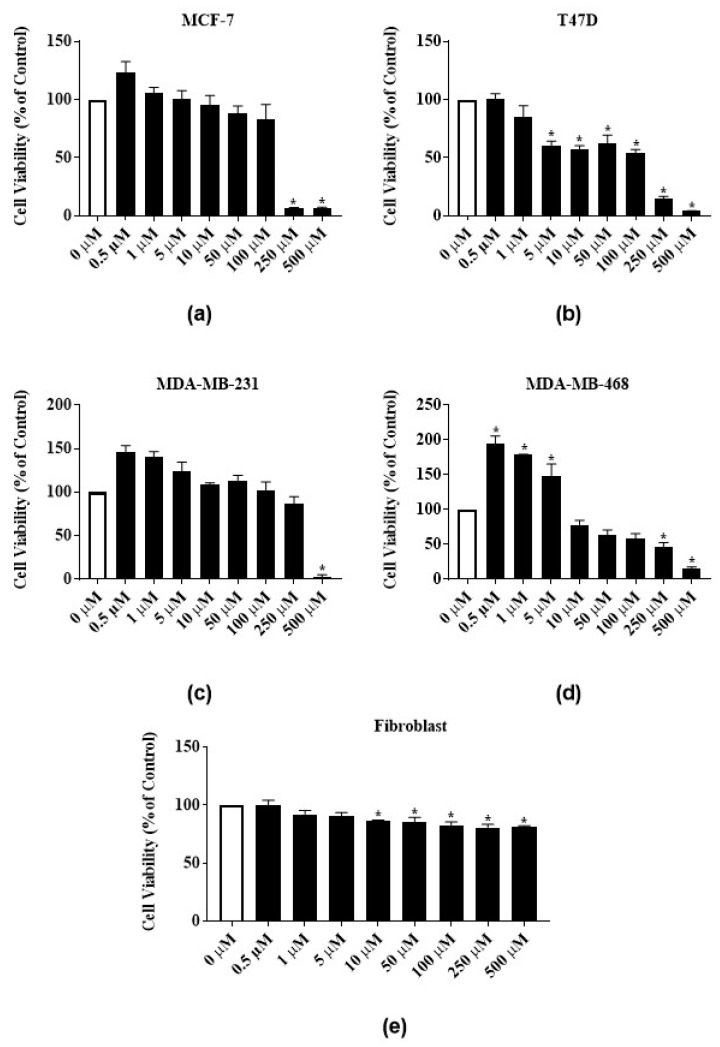
Antiproliferative activities of *E*. *bicolor* latex phytochemical RTX on ER-positive and triple negative breast cancer cell lines: (**a**) MCF-7 (ER+); (**b**) T47-D (ER+); (**c**) MDA-MB-231 (triple negative); (**d**) MDA-MB-468 (triple negative) breast cancer cell lines; and (**e**) effects of RTX on the growth of human normal primary dermal fibroblast. Bars with asterisks are significantly different from untreated control at *p* ≤ 0.05 (One-way ANOVA followed by Tukey’s posthoc test).

**Figure 5 nutrients-12-00059-f005:**
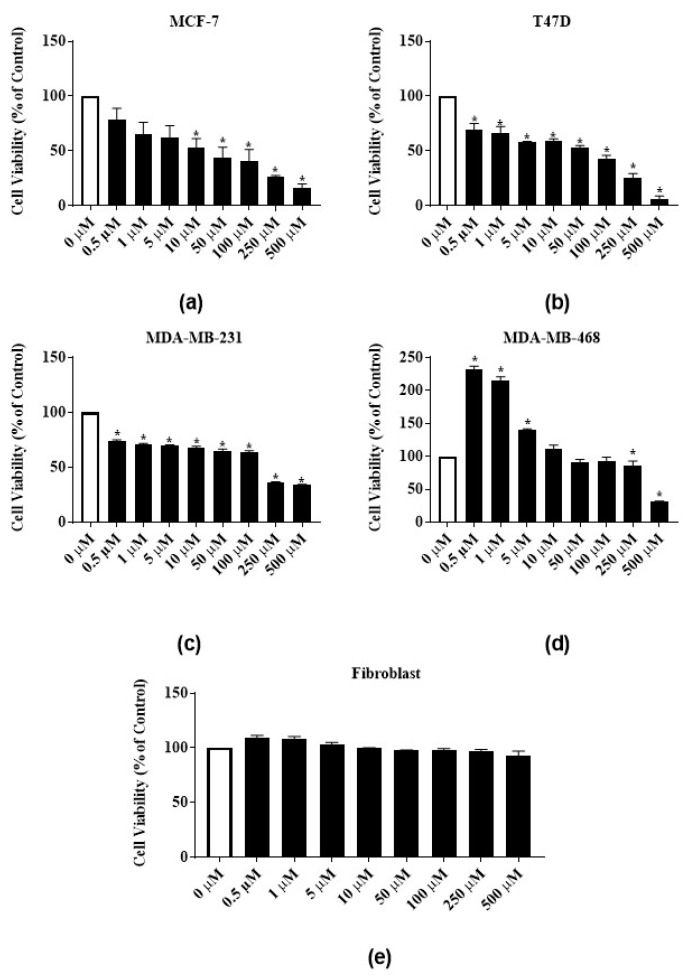
Antiproliferative activities of *E*. *bicolor* latex phytochemical rutin on ER-positive and triple negative breast cancer cell lines: (**a**) MCF-7 (ER+); (**b**) T47-D (ER+); (**c**) MDA-MB-231 (triple negative); (**d**) MDA-MB-468 (triple negative) breast cancer cell lines; and (**e**) effects of RTX on the growth of human normal primary dermal fibroblast (**e**). Bars with asterisks are significantly different from untreated control at *p* ≤ 0.05 (One-way ANOVA followed by Tukey’s posthoc test).

**Figure 6 nutrients-12-00059-f006:**
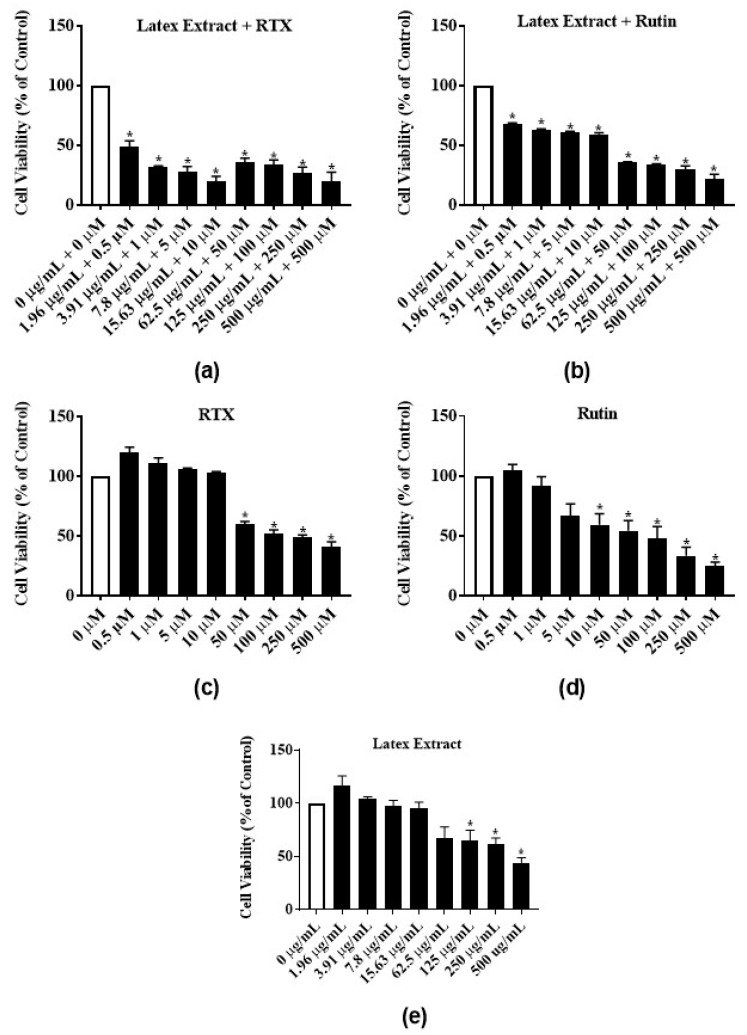
Antiproliferative activities of *E*. *bicolor* latex extract in combination with RTX or rutin on MCF-7 breast cancer cell line: (**a**) *E*. *bicolor* extract + RTX; (**b**) latex extract + rutin; (**c**) RTX; (**d**) rutin (**e**) latex extract. Cell treatment with the combination latex extract-phytochemical significantly reduced the proliferation of MCF-7 cells as compared to the corresponding individual treatments of only phytochemicals or latex extract. Bars with asterisks are significantly different from untreated control at *p* ≤ 0.05 (One-way ANOVA followed by Tukey’s posthoc test).

**Table 1 nutrients-12-00059-t001:** The growth inhibition 50 (GI_50_) of *E*. *bicolor* latex extract and its phytochemicals RTX and rutin in MCF-7 and T47D (ER+) and MDA-MB-231 and MDA-MB-468 (triple negative) breast carcinomas.

***E*. *bicolor* Latex Extract**	
**Breast Carcinoma**	**GI_50_ (µg/mL)**
MCF-7	498.7 ± 1.3
T47D	315.7 ± 36.6
MDA-MB-231	258.3 ± 18
MDA-MB-468	499 ± 0.8
**RTX**	
**Breast Carcinoma**	**GI_50_ (µM)**
MCF-7	139 ± 7.8
T47D	100 ± 23.6
MDA-MB-231	246.7 ± 3.4
MDA-MB-468	248.5 ± 1.5
**Rutin**	
**Breast Carcinoma**	**GI_50_ (µM)**
MCF-7	77.5 ± 18.8
T47D	65.7 ± 14
MDA-MB-231	160 ± 8.2
MDA-MB-468	383.3 ± 54.4

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
