# Peer review of "Estrogenic, Antiestrogenic and Antiproliferative Activities of Euphorbia bicolor (Euphorbiaceae) Latex Extracts and Its Phytochemicals"

_nutrients, 2019, doi:10.3390/nu12010059_

Round 1

Reviewer 1 Report

In the present study the authors tested the ability of the extracts from Euphorbia bicolar for its estrogenic, anti-estrogenic and anti-proliferative effects in breast cancer cell lines. They found that Coumestrol and Genistein exhibited higher estrogenic and anti-estrogenic activities compared to diterpenes and other flavonoids. Interestingly RTX and rutin induced antiproliferative activities only in cancer cell lines in a dose dependent fashion and was not observed in primary fibroblast cultures. The effect of the latex extract was biphasic in MDA-MB-436 cells with lower concentrations potentiating the growth and higher concentrations inhibiting growth. Additional findings of the study include the observation that at lower concentrations latex extract with rutin/RTX had a synergistic inhibitory effect on cell proliferation while the effect was determined to be additive at higher concentrations.

Minor comments:

Please explain why fibroblasts were used instead of primary breast epithelial cells. In line 287, please correct the spelling of treatment. Currently it reads a “reatment”. Figure 3 and some panels in Figure 6 ae in μg/ml whereas in other figures (Fig. 5 and 4 and other panels of Fig 6) concentration is represented in μ We suggest the authors to be consistent in labeling the concentrations (i.e.) either use μg/ml or μM throughout the manuscript. The finding that the extracts containing flavonoids have anti-proliferative effects is very interesting. Since flavonoids are unstable compounds, it is likely that the observed effects is related to their degraded products rather than the parent compounds. In this context, there is a paper that showed flavonoid metabolites have antiproliferative effects (even in MDA-MB-231 cells). “Sankaranarayanan, R.; Valiveti, C.K.; Kumar, D.R.; Van slambrouck, S.; Kesharwani, S.S.; Seefeldt, T.; Scaria, J.; Tummala, H.; Bhat, G.J. The Flavonoid Metabolite 2,4,6-Trihydroxybenzoic Acid Is a CDK Inhibitor and an Anti-Proliferative Agent: A Potential Role in Cancer Prevention. Cancers 2019, 11, 427.” The authors should discuss their results in the context of this finding.

Reviewer 2 Report

Dear Authors,

This work is interesting and very well presented.

I would suggest a few adds:

If possible it would be interesting to have more informations on the content of active compounds in the plant (evaluation) and in the extract and relative % of the compouds in the extract (maybe showing chromatograms).

Also I suggest to enlight the agonistic/antagonistic effects of genistein (and other compounds with same behavior) at low vs. high concentrations with a comparison with clinical studies when available including in relation with diet).

On the comparison of the efficacy of the differents Euphorbia species  please describe the difference of composition and or relative % of main active compounds.

Best regards
